# Outpatient antibiotic prescription rate and pattern in the private sector in India: Evidence from medical audit data

**Habib Hasan Farooqui** [1]*, **Aashna Mehta**[2], **Sakthivel Selvaraj**[2]

**1** Indian Institute of Public Health –Delhi, Public Health Foundation of India, Gurugram, Haryana, India,
**2** Health Economics, Financing and Policy, Public Health Foundation of India, Gurugram, Haryana, India

* drhabibhasan@gmail.com

**Data Availability Statement:** The data underlying the results presented in the study are available from the IMS Health (now IQVIA). The IMS Healtth can be approached for data access through their website (https://www.iqvia.com/locations/india).

## Abstract

The key objective of this research was to generate new evidence on outpatient antibiotic prescription rate and patterns in the private sector in India. We used 12-month period (May 2013 to April 2014) medical audit dataset from IQVIA (formerly IMS Health). We coded the diagnosis provided in the medical audit data to International Statistical Classification of Diseases and Related Health Problems (ICD-10) and the prescribed antibiotics for the diagnosis to Anatomic Therapeutic Chemical (ATC) classification of World Health Organization (ATC index-2016). We calculated and reported antibiotic prescription rate per 1,000 persons per year, by age groups, antibiotic class and disease conditions. Our main findings are—approximately 519 million antibiotic prescriptions were dispensed in the private sector, which translates into 412 prescriptions per 1,000 persons per year. Majority of the antibiotic prescriptions were dispensed for acute upper respiratory infections (J06) (20.4%); unspecified acute lower respiratory infection (J22) (12.8%); disorders of urinary system (N39) (6.0%); cough (R05) (4.7%); and acute nasopharyngitis (J00) (4.6%) and highest antibiotic prescription rates were observed in the age group 0–4 years. To conclude our study reports first ever country level estimates of antibiotic prescription by antibiotic classes, age groups, and ICD-10 mapped disease conditions.

## Introduction

India is considered to be one of the top users of antibiotics. Our previous research had reported that per capita antibiotic consumption in the retail sector in India has increased by around 22%, from 13.1 DID (defined daily dose (DDD) per 1000 inhabitants per day) in 2008 to 16.0 DID, in a span of five years (2012 to 2016).[1] Evidence from another study suggests that between 2000 and 2015, antibiotic consumption increased from 3.2 to 6.5 billion DDDs (103%) while the antibiotic consumption rate increased from 8.2 to 13.6 DIDs (63%) in India. [2] Literature suggests that high burden of infectious diseases could be one of the reasons for high antibiotic use in India. As per the Million Death Study (MDS) diseases of infectious origin such as pneumonia and diarrhea accounted for around 50% (0·67 million of 1·34 million) of

**Funding:** Habib Hasan Farooqui is supported by Department of Science and Technology's Public Health Research Grant to the Public Health Foundation of India. The funders had no role in study design, data collection and analysis, decision to publish, or preparation of the manuscript.

**Competing interests:** The authors have declared that no competing interests exist.

all deaths in children aged less than 5 years in India.[3] However, inappropriate use of antibiotics cannot be ruled out.

Although clinical guidelines on judicious antibiotic use[4, 5] explicitly mentions that antibiotics should not be prescribed for common cold, nonspecific upper respiratory tract infection (URI), acute cough illness, and acute bronchitis, literature on antibiotic prescribing from India indicates high rate of antibiotic prescriptions for respiratory infections in primary care.[6–8]. Recent evidence from United States also suggests that seasonal peaks in antibiotics use during cold and influenza season suggestive of viral upper respiratory tract infections antibiotic has remained unchanged highlighting inappropriate prescribing of antibiotics [9]despite release of several antibiotic prescribing guidelines. However, evidence from UK suggest that standardized consultation rate for 'any respiratory infection' declined by 35 per cent and overall antibiotic prescriptions for all acute respiratory infections declined by 45 per cent between 1994 and 2000.[10] Another interesting study from Taiwan reported that children with a physician or a pharmacist as a parent were significantly less likely than other children to receive antibiotic prescriptions suggesting better education does help in reducing the frequency of injudicious antibiotic prescribing.[11]

Evidence from India has also highlighted relatively high antibiotic prescription rate in private health facilities as compared to public health facilities.[7, 12] These studies also highlighted the frequent use of expensive newer classes of antibiotics as compared to the older ones in the private sector.[7, 8, 12, 13] One of the possible reasons for such a trend is the dominance of the private sector in funding and provisioning of health care in India, as per the National Sample Survey (NSS) nearly 75% of all outpatient visits and about 62% of hospitalization episodes occurred in private health delivery system in the year 2014.[14] Furthermore, households largely buy medicines directly from retail pharmacies as prescribed by the general practitioners in the private sector.[15]

The health system elements outlined above clearly demonstrate the role and relevance of primary care physician/general practitioner of the private sector in medicine use and in particular antibiotic use. Previous research involving micro level surveys revealed several facets of inappropriate medicine use in the Indian context.[7, 8, 16–18] However, none of them were truly representative of private sector primary care physicians because of their limited sample size and limited geographical locations. We conducted this research to generate new evidence on outpatient antibiotic (J01) prescription rate and patterns in the private sector in India. We also performed additional analysis to report age-specific and disease-specific differences by different antibiotic classes.

## Material and methods

### Data source and setting

We examined prescription rates and patterns of antibiotics (J01) of primary care physicians working in the private sector in India with the help of IQVIA medical audit data (formerly IMS Health) for a 12 month period (May 2013 to April 2014).[19] IQVIA is a for-profit organisation that collects and provides data and information on pharmaceutical market intelligence in over 100 countries around the world. The medical audit data tracks prescriptions by private practitioners practicing allopathic system of medicine. This data is collected from a panel comprising of 4600 doctors selected through a multi-stage stratified random sampling, which include general practitioners (MBBS, Bachelor of Medicine, Bachelor of Surgery), non-MBBS general practitioners, and other medical specialties (such as dentists, pediatrics, gynecology, dermatology, and others) from 23 metropolitan areas (population more than 1 million), 128 class 1 towns (population more than 100,000) and 1A towns (population less than 100,000) of

India. The data is then extrapolated to reflect the prescription pattern of doctors having private practices in towns with population more than a hundred thousand across the country.

This database provides information on patient characteristics such as age, gender, symptoms, diagnosis and the medicines prescribed. The data organizes medicines according to anatomical therapeutic classification (ATC) of the European Pharmaceutical Market Research Association (EphMRA) but not according to the World Health Organisation's ATC classification. Also, the diagnosis reported on prescriptions are not coded to the International Statistical Classification of Diseases and Related Health Problems (ICD-10). Furthermore, the data does not capture the public sector prescriptions and therefore our analysis only reflects outpatient antibiotic prescription patterns in the private sector in the country.

Finally, the data was made available to us by IQVIA in the form of aggregates processed and extrapolated to reflect the prescription practices in the country instead of being made available in raw form as individual-level data. The data we used had no identifiers for the patients. We therefore did not require ethical approval for our study.

## Outcome measure

Our primary outcome measure was antibiotic prescription rate per 1,000 persons per year. We also estimated and reported age-specific and disease-specific antibiotic prescriptions by antibiotic classes.

## Statistical analysis

We coded the diagnosis provided on the prescription in the IQVIA medical audit data to disease classifications based on the International Statistical Classification of Diseases and Related Health Problems, 10th Revision (ICD-10 classification; version: 2016)[20] and the antibiotics prescribed for the related diagnosis to the 3rd level of Anatomic Therapeutic Chemical (ATC) classification as per the methodology proposed by World Health Organization's Collaborating Centre (WHOCC) of Drug Statistics Methodology's (ATC index-2016).[21] The diagnosis was categorized into ICD 10 codes through a search on the online index using specific key words in the diagnosis provided in the medical audit data, which was taken directly from the prescriptions. The idea was to code the ICD 10 codes up to the narrowest (most detailed) level possible depending on the extent of details on the diagnosis provided in the medical audit data.

The utilization of antibiotics (ATC code: J01) was measured in terms of the annual prescription rate, i.e. number of antibiotic prescriptions divided by 1000 person years. The population estimates were obtained from the report of the technical group on population projections constituted by the National Commission on Population.[22] Age-groups were determined by the classification already provided in the medical audit data. The medicines prescribed were classified into the following antibiotic subgroups (ATC codes): tetracyclines (J01A), amphenicols (J01B), penicillins (J01C), other beta-lactams, cephalosporins (J01D), sulfonamides & trimethoprim (J01E), macrolides, lincosamides and streptogramins (J01F), aminoglycosides (J01G), Quinolones (J01M), combinations of antibacterials (J01R), other antibacterials (J01X). Unclassifiable antibiotics were pooled in the subgroup 'others'.

We used per 1000 persons as denominator in contrast to individuals because the data was available only for prescription per person and not for an individual for the entire year. Antibiotic prescription rate is a better indicator for antibiotic use[23] as compared to defined daily doses (DDD) per person since antibiotic dose depends on a patient's age and body weight. We analyzed and reported antibiotic use by age-groups and disease conditions (based on ICD-10 classification) expressed as annual prescription rate per 1,000 persons for each class of antibiotics. We used software STATA 14.0 to perform statistical analysis.

## Results

### Antibiotic prescribing pattern

We present new evidence on outpatient antibiotic prescription rates and pattern in the private sector in India. Around 519 million antibiotic prescriptions were dispensed in 2014, which translate into 412 prescriptions per 1,000 persons per year. The antibiotic prescription rates were highest for children aged 0–4 years (636 prescriptions per 1,000 persons) and lowest in the age group 10–19 years (280 prescriptions per 1,000 persons) (Fig 1 and S1 Table). It may also be noted that across all age groups, beta-lactam, cephalosporins (J01D) had the highest prescription rates (38.3% of all antibiotic prescriptions) followed by beta-lactam, penicillins (J01C) (22.8%) and quinolones (J01M) (16.3%). It may be further noted that cephalosporins (J01D) were the most commonly prescribed antibiotic across all diagnoses with the exception of disorders of urinary system where quinolones (J01M) were more commonly prescribed.

### Antibiotic prescribing across clinical diagnoses

Of 519 million antibiotic prescriptions, majority were dispensed for the diseases of the respiratory system (55%), followed by diseases of the genitourinary system (10%), and symptoms, signs and abnormal clinical findings (9%) (Table 1).

As per the ICD-10 classification, the following top ten disease conditions contributed approximately 63% of the total antibiotic prescriptions—acute upper respiratory infections (J06) (20.4%; range—12.1% to 25.4%), unspecified acute lower respiratory infection (J22) (12.8%; range—9.0% to 24.7%), disorders of urinary system (N39) (6.0%; range—1.7% to 9.0%), cough (R05) (4.7%; range—2.9% to 6.1%), acute nasopharyngitis (J00) (4.6%; range— 2.0% to 6.1%), acute pharyngitis (J02) (3.9%; range– 1.9% to 5.0%), acute bronchitis (J20) (3.4%; range—2.4% to 5.1%), injury, poisoning and others (T14) (2.5%; range– 1.8% to 3.2%), cutaneous abscess and furuncle (L02) (2.3%; range—1.4% to 2.6%), and asthma (J45) (2.2%; range—1.8% to 4.4%) (Fig 2 and S2 Table). (For full information on antibiotic prescription percentages across top ten clinical diagnosis and age groups and across antibiotic classes see supplementary file (S1, S2 and S3 Tables)).

## Discussion

To the best of our knowledge, our study provides the first ever estimates of outpatient antibiotic prescription rates and patterns in the private sector in towns with population more than a hundred thousand across the country. Our findings illustrate significant variations in antibiotic prescription rates across age groups, by disease conditions (ICD-10 classification) and by antibiotic classes (ATC classification). Earlier Von Bockel et al. and Klein et al. had reported estimates of antibiotic consumption in India through use of pharmaceutical sales data[2, 24]. However, their study did not provide information on antibiotic use by disease conditions and age groups. While other prescription analysis based studies from India reported antibiotic utilization in public and private sector health facilities[7, 8, 12, 13, 16, 25], they had limited sample sizes and geographical locations.

Our estimates suggest high proportion of antibiotic prescription for upper respiratory tract infections (acute upper respiratory infections (J06) (20.4%), cough (R05) (4.7%), acute nasopharyngitis (J00) (4.6%), and acute pharyngitis (J02) (3.9%)). Generally, these infections are viral in origin and are self-limiting in nature. Hence, in the light of evidence based medicine and standard treatment guidelines, it may be argued that a significant proportion of these antibiotic prescriptions might be inappropriate in nature. Previous research on prescription practices also highlighted the problem of inappropriate use of broad-spectrum antibiotics in India.

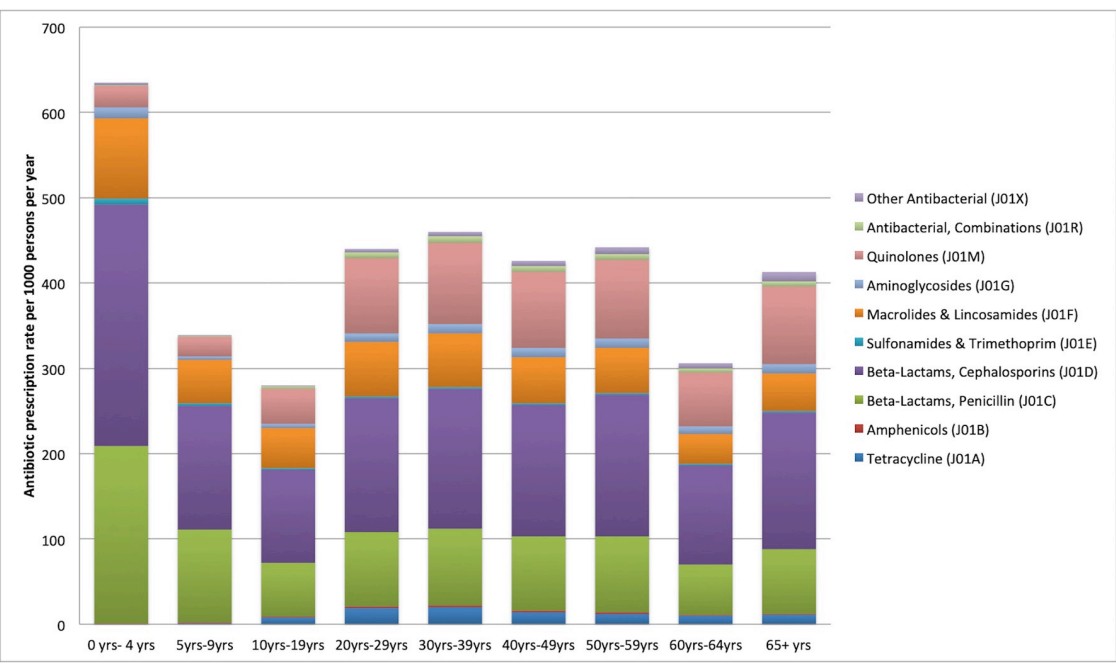

**Fig 1. Outpatient antibiotic prescription rate per 1000 persons per year, by age groups and antibiotic classes, India (2013–2014).**

**Table 1. Distribution of outpatient antibiotic prescriptions in India, by disease conditions, 2013–2014.**

| ICD chapter number | ICD chapter name | Number of prescriptions | Percentage (%) | Prescription rate per 1000 persons per year |
|---|---|---|---|---|
| 1 | Certain infectious and parasitic diseases | 16,820,293 | 3.24 | 13.35 |
| 3 | Diseases of the blood and blood-forming organs | 1,158,125 | 0.22 | 0.92 |
| 4 | Endocrine, nutritional and metabolic diseases | 4,566,575 | 0.88 | 3.63 |
| 5 | Mental and behavioral disorders | 967,392 | 0.19 | 0.77 |
| 6 | Diseases of the nervous system | 1,491,169 | 0.29 | 1.18 |
| 7 | Diseases of the eye and adnexa | 3,685,540 | 0.71 | 2.93 |
| 8 | Diseases of the ear and mastoid process | 8,653,317 | 1.67 | 6.87 |
| 9 | Diseases of the circulatory system | 8,472,213 | 1.63 | 6.73 |
| 10 | Diseases of the respiratory system | 286,059,212 | 55.09 | 227.10 |
| 11 | Diseases of the digestive system | 28,532,251 | 5.49 | 22.65 |
| 12 | Diseases of the skin and subcutaneous tissue | 20,917,567 | 4.03 | 16.61 |
| 13 | Diseases of the musculoskeletal system | 5,973,066 | 1.15 | 4.74 |
| 14 | Diseases of the genitourinary system | 51,791,862 | 9.97 | 41.12 |
| 15 | Pregnancy, childbirth and the puerperium | 977,532 | 0.19 | 0.78 |
| 17 | Congenital malformations, deformations | 114,892 | 0.02 | 0.09 |
| 18 | Symptoms, signs and abnormal clinical and laboratory | 44,506,708 | 8.57 | 35.33 |
| 19 | Injury, poisoning and external causes | 33,398,837 | 6.43 | 26.52 |
| 20 | External causes of morbidity and mortality | 37,318 | 0.01 | 0.03 |
| 21 | Factors influencing health status | 1,118,282 | 0.22 | 0.89 |
| | Total | 519,242,151 | 100.00 | 412.23 |

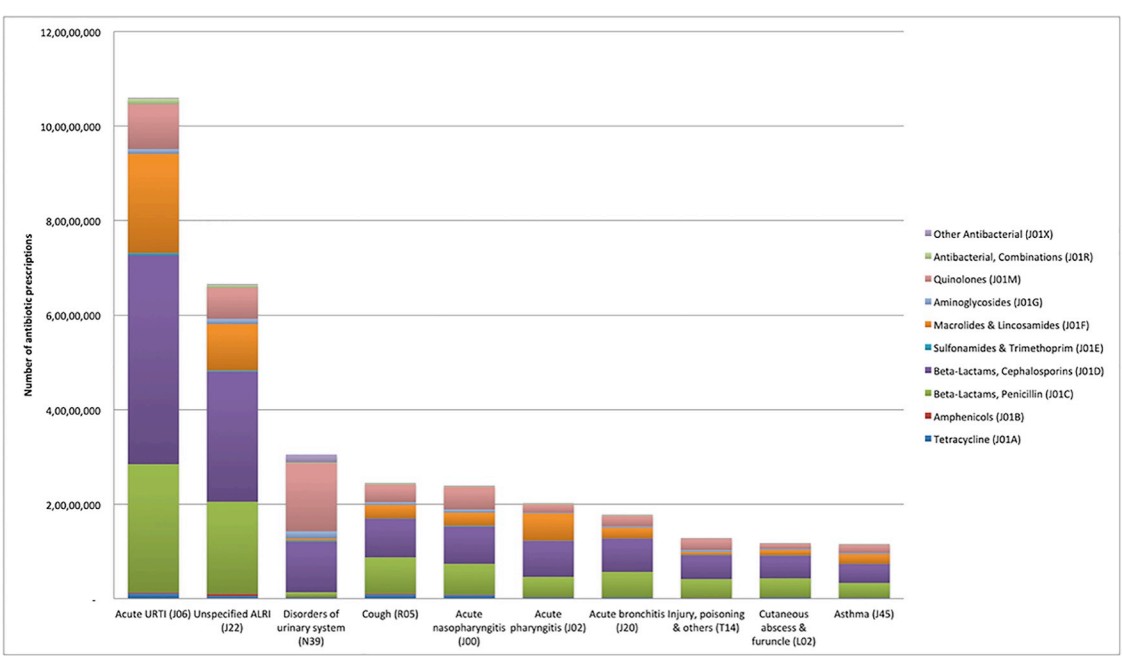

**Fig 2. Number of antibiotic prescriptions, by disease conditions and antibiotic classes, India (2013–2014).**

For example, Kotwani et al. had reported that in the private sector, not only were the antibiotic prescription rates higher but the choice of antibiotics for the treatment of uncomplicated respiratory infections too, was inappropriate.[8] Chandy et al reported widespread use of fluoroquinolone, especially by general practitioners[12] in the private sector. Other studies have also reported high cephalosporin use in urban hospitals and pharmacy shops.[12, 26]

Bianco et al, who studies antibiotic prescription to adults with ARTI by Italian GPs, concluded that there was a very high frequency of non evidence-based prescriptions of antibiotics at primary care level. 65.5 percent of times the prescriptions were not being indicated by guideline. [27]Studies have also highlighted the incorrect perception among patients and parents of pediatric patient that antibiotics work for treating viral infections.[28, 29]Napolitano et al reported that only 9.8 percent of the Italian patients surveyed knew the definition of antibiotic resistance and only 21.2 percent knew when it was appropriate to use antibiotics. [30]

Previous research also shows that besides the lack of awareness, inappropriate antibiotic use is linked to supply-side incentives, which lead to over prescription of antibiotics in the private sector.[16, 25, 31] This problem of inappropriate use of antibiotics gets accentuated multifold because of limited access to care and medicines in the public health system[32] which forces patients to seek care in the private sector. Limited access to medicines in the public sector also results in over the counter purchase of antibiotics, which is a major driver of inappropriate use in India. Laxminarayanan et al had reported that non-prescription sales of carbapenems in India are among the highest in the world and contribute to growing carbapenem resistance. [33] Although, over the counter access to antibiotics is a complex problem in India since insufficient access and delays in access to antibiotics causes more deaths than antibiotic resistance., [34] Numerous studies have reported increasing levels of resistance to last resort antibiotics like carbapenem[33, 35, 36]. Evidence also suggests that inappropriate antibiotic use not only has profound impact on antimicrobial resistance[37] but also on treatment cost because of drug resistant organism.[38]

Our analyses suggest that antibiotic prescription rates (412 prescriptions per 1000 persons in 2014) in India are lower than various European nations. For example, antibiotic prescription rates in Italy (957 prescriptions per 1000 persons), Germany (561 prescriptions per 1000 persons), UK (555 prescriptions per 1000 persons), Denmark (481 prescriptions per 1000 persons)[39] and Greece (1100 antibiotics per 1000 person)[26] are much higher than India. However, antibiotic prescription rates for certain antibiotic classes are on a higher side in India as compared to the developed nations. For example, the percentage of prescriptions with cephalosporins and quinolones (38.2% and 16.3%) in India were significantly higher than USA (14.0% and 12.7%)[40], and Greece (32.9% and 0.5%).[26] Such unusually high prescription rates of beta-lactams-penicillins and cephalosporins in uncomplicated upper respiratory infections in children is in stark contrast to the prescription rates and pattern reported in European countries.[39]

The potential reasons for high prescription rates of broad spectrum antibiotics like cephalosporins and quinolones are not only high burden of infectious diseases, but also lack of diagnostic support services and inadequate training of physicians.[12] Literature also suggests that perceived demand and expectations from the patients, influence from medical representatives and inadequate knowledge influences doctors decisions to prescribe antibiotics.[41] Previous research has highlighted that inappropriate use of medicines is rampant in less regulated health markets[42] and can take several forms: overuse, underuse, misuse, and unnecessary expensive use. Our findings for antibiotic prescription are consistent with antibiotic sales in India. Our previous analysis on pharmaceutical sales data had also suggested that antibiotic consumption (16.0 DID) in India was significantly below the mean antibiotic consumption (21.5 DID) of European countries.[1]

To address the problems related to inappropriate use of antibiotics, government of India has deployed a multipronged strategy. This includes setting up treatment guidelines for antimicrobial use[5] and multi-centric surveillance for tracking antimicrobial resistance[43]. In addition, over-the-counter (OTC) sales of 3rd and 4th generation antibiotics are now regulated through Schedule H1 of Drugs & Cosmetics Rules.[44] Furthermore, to reduce burden of pneumonia and diarrhea and the demand for antibiotics, new vaccines have been introduced in the universal immunization program[45]. However, success of these strategic measures in terms of achieving intended objectives is yet to be demonstrated. Our analysis suggest around 100 million prescriptions were dispensed for acute upper respiratory tract infections alone, antibiotic stewardship programs directed towards diagnosis and treatment of URI could result into significant reduction in antibiotic use.

Our study has certain limitations. The scope of our study was limited to analysis of antibiotic prescription pattern in the private sector in in towns with population more than a hundred thousand across the country, as we did not have access to public sector prescription data. Therefore, the study is not representative of the prescriptions generated in public sector facilities. This may have resulted in an underestimation of antibiotic prescription rates albeit only marginally, since more than 80% of the population seeks care in private sector and approximately 90% of medicine expenditure occurs in private sector. In addition, a small proportion (<1%) of antibiotic prescriptions sometimes got mapped to completely unrelated diagnosis because IQVIA medical audit data is coded in such a way that antibiotics on prescriptions gets mapped to every differential diagnosis (related or completely unrelated) on the prescription. The diagnosis provided in the dataset were not already mapped to ICD 10 classification, therefore the authors had to map the diagnosis to ICD 10 classification based on available information which may have led to certain inaccuracies in allocation of codes as well as the resulting analysis. Since we did not have information on the number of patients accessing general

practices in metros and class 1 and 1a towns, we had to rely on the total population to arrive at the prescription rates. This is another limitation of our study.

## Conclusions

This research work provides the first ever estimates on antibiotic prescription rate and pattern in the outpatient general practice in the private sector of towns with population more than a hundred thousand across the country. Overall antibiotic prescription rates in India are still much lower than Europe. However, prescription rates for broad-spectrum beta-lactam antibiotics are much higher as compared to European nations, especially in children. Approximately one-fifth antibiotic prescriptions were dispensed for upper respiratory infections, which rarely require an antibiotic therapy. Our findings highlight that primary care physicians in the private sector can play a key role in reducing antibiotic misuse and overuse. Our research findings also provide critical information to target antimicrobial stewardship programs to specific constituencies and stakeholders. This baseline information can also be used as a benchmark for measuring the impact of current and future interventions directed towards reducing inappropriate antibiotic use.

## Supporting information

**S1 Table. Antibiotic prescription rate per 1000 person per year, by age group, India (2013–2014).**
(DOCX)

**S2 Table. Top 10 diagnoses for antibiotic prescriptions, by age group, India (2013–2014).**
(DOCX)

**S3 Table. Distribution of outpatient antibiotic prescriptions by disease conditions and antibiotic classes, India (2013–2014).**
(DOCX)

## Acknowledgments

The authors would like to thank Ms. Manushi Sharma for the data coding support.

## Author Contributions

**Conceptualization:** Habib Hasan Farooqui, Sakthivel Selvaraj.

**Data curation:** Aashna Mehta.

**Formal analysis:** Aashna Mehta.

**Funding acquisition:** Habib Hasan Farooqui.

**Methodology:** Habib Hasan Farooqui, Aashna Mehta, Sakthivel Selvaraj.

**Resources:** Sakthivel Selvaraj.

**Supervision:** Habib Hasan Farooqui, Sakthivel Selvaraj.

**Writing – original draft:** Habib Hasan Farooqui.

**Writing – review & editing:** Habib Hasan Farooqui, Aashna Mehta, Sakthivel Selvaraj.

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
