## [Decision Letter · Decision Letter 0]

21 Jun 2019

PONE-D-19-14711

Outpatient antibiotic prescription rate and pattern in the private sector in India: Evidence from medical audit data

PLOS ONE

Dear Dr. Farooqui,

Thank you for submitting your manuscript to PLOS ONE. After careful consideration, we feel that it has merit but does not fully meet PLOS ONE’s publication criteria as it currently stands. Therefore, we invite you to submit a revised version of the manuscript that addresses the points raised during the review process.

We would appreciate receiving your revised manuscript by July 23, 2019. To enhance the reproducibility of your results, we recommend that if applicable you deposit your laboratory protocols in protocols.io, where a protocol can be assigned its own identifier (DOI) such that it can be cited independently in the future. For instructions see: http://journals.plos.org/plosone/s/submission-guidelines#loc-laboratory-protocols

We look forward to receiving your revised manuscript.

Kind regards,

Italo Francesco Angelillo, DDS, MPH

Academic Editor

PLOS ONE

Journal Requirements:

2. In ethics statement in the manuscript and in the online submission form, please provide additional information about the patient records used in your retrospective study. Specifically, please ensure that you have discussed whether all data were fully anonymized before you accessed them and/or whether the IRB or ethics committee waived the requirement for informed consent. If patients provided informed written consent to have data from their medical records used in research, please include this information.

3.  We noticed your Discussion has some minor occurrence of overlapping text with the following previous publication, which needs to be addressed:

Farooqui, Habib Hasan, et al. "Community level antibiotic utilization in India and its comparison vis-à-vis European countries: Evidence from pharmaceutical sales data." PloS one 13.10 (2018): e0204805.

In your revision ensure you cite all your sources (including your own works), and quote or rephrase any duplicated text outside the methods section. Further consideration is dependent on these concerns being addressed.

Reviewers' comments:

Reviewer's Responses to Questions

**Comments to the Author**

1. Is the manuscript technically sound, and do the data support the conclusions?

Reviewer #1: Yes

Reviewer #2: Partly

2. Has the statistical analysis been performed appropriately and rigorously? 

Reviewer #1: I Don't Know

Reviewer #2: Yes

3. Have the authors made all data underlying the findings in their manuscript fully available?

Reviewer #1: No

Reviewer #2: Yes

4. Is the manuscript presented in an intelligible fashion and written in standard English?

Reviewer #1: Yes

Reviewer #2: Yes

5. Review Comments to the Author

Reviewer #1: Dr. Farooqui and colleagues present a descriptive analysis of private-sector antibiotic prescribing in India using IQVIA medical audit data from May 2013-May 2014. The study presents descriptive findings that might be of interest to clinicians and public health practitioners. I felt the analysis did not go very deep into the data, although the discussion nicely highlighted key findings. The manuscript could benefit from a deeper analysis of the data, if possible given the dataset. A few comments for consideration:

• As the diagnoses in your dataset are not in ICD10 codes, how did you categorize them according to ICD10 codes? It would be helpful to include more on this methodology, perhaps consider including a crosswalk in your supplementary materials.

• Why are there no variance estimates? Is this due to the dataset projection methodology? If possible, variance estimates should be included.

• It would make your manuscript stronger if you dug deeper into the descriptive data with a few additional analyses. For example, what diagnoses are responsible for the most antibiotic prescriptions by age group? Is there a statistically significant difference between agents and diagnoses in different age groups.

• Is there any information on region/geographic area in the dataset. Or provider type (more specifically than general practitioner)? That could be interesting to include and might make the analyses more robust.

• What does the ICD code column in Table 1 show?

• In the discussion, it makes sense that you discuss over the counter antibiotic sales as that is a contributor to inappropriate antibiotic use. However, I feel you could trim this section down a little since it is not the focus of your analysis.

• Please review for grammar and punctuation and ensure all abbreviations are defined at their first use.

• I think it would be fine to say IQVIA instead of IMS Health since that is the current company name.

Reviewer #2: Although this approach is meriting, the issues of the paper are not meaningful enough and convincing. I do not agree with the conclusion of the authors: data results could not lead to the conclusion “We observed an inappropriate and high antibiotic prescription rates

for upper respiratory infections in children age less than 5 years”. Indeed, you used a dataset to estimate the rates and the motif of prescription; to prescribe an antibiotic is associated with a lot of arguments and to estimate if appropriate or not you must read the entire medical report, with biological results (you did not include) the symptoms, the X-rays and so on. To use retrospective medical and clinical data to estimate the rate of AB prescription is a tool we have to develop nowadays, but you can’t conclude on pertinence with this kind of methods.

There are different potential methodological reasons for that, the main being that very different clinical presentations could lead to quote a upper respiratory infections; but if the person has comorbidities or other medical condition, maybe an antibiotic could be necessary. You did not adjust your results on the conditions of the patients. The study could be more relevant in presenting the results without this kind of interpretation that can’t be done.

6. PLOS authors have the option to publish the peer review history of their article (what does this mean?). If published, this will include your full peer review and any attached files.

Reviewer #1: No

Reviewer #2: No

---

## [Author Response · Author response to Decision Letter 0]

16 Sep 2019

Outpatient antibiotic prescription rate and pattern in the private sector in India: Evidence from medical audit data

Response to Reviewers 

The authors would like to express their gratitude to the reviewers for their insightful comments that have been extremely useful for us in improving the manuscript.

Reviewer #1: 

Dr. Farooqui and colleagues present a descriptive analysis of private-sector antibiotic prescribing in India using IQVIA medical audit data from May 2013-May 2014. The study presents descriptive findings that might be of interest to clinicians and public health practitioners. I felt the analysis did not go very deep into the data, although the discussion nicely highlighted key findings. The manuscript could benefit from a deeper analysis of the data, if possible given the dataset. 

A few comments for consideration:

As the diagnoses in your dataset are not in ICD10 codes, how did you categorize them according to ICD10 codes? It would be helpful to include more on this methodology, perhaps consider including a crosswalk in your supplementary materials.

Response: The following has been included in the section ‘Material and Methods’ sub-section ‘Statistical Analysis’ Para 1, page 6 of the manuscript to explain the coding process better:

“The diagnosis was categorized into ICD 10 codes through a search on the online index using specific key words in the diagnosis provided in the dataset which was taken directly from the prescriptions. The idea was to code the ICD 10 codes up to the narrowest (most detailed) level possible depending on the extent of details provided in the diagnosis.”

Reviewer: Why are there no variance estimates? Is this due to the dataset projection methodology? If possible, variance estimates should be included.

Response: We worked with data that was already projected and shared with us by IQVIA (formerly IMS Health). We did not carry out the projection ourselves. We are therefore unable to provide variance estimates. 

Reviewer: It would make your manuscript stronger if you dug deeper into the descriptive data with a few additional analyses. For example, what diagnoses are responsible for the most antibiotic prescriptions by age group? Is there a statistically significant difference between agents and diagnoses in different age groups.

Response: Thank you for the valuable suggestion. We have now included the table with the top 10 diagnosis for antibiotic prescriptions by age groups in the supplementary material. Please see S2 Table in supplementary materials. This has been referred to in the ‘results’ sections, sub-section ‘Antibiotic prescribing across clinical diagnoses’ para 1, page 10-11. 

Reviewer : Is there any information on region/geographic area in the dataset. Or provider type (more specifically than general practitioner)? That could be interesting to include and might make the analyses more robust.

Response: Unfortunately, this information is not available with us. It must be noted that while the data is from private practitioners, it is not only from GPs but also representative of other specializations. We do not however, have information on how the prescription patterns vary across prescriber specializations. 

Reviewer : What does the ICD code column in Table 1 show?

Response: The ICD code indicates each of the 21 chapters. This has been re-labeled in table 1 in the manuscript. 

Reviewer : In the discussion, it makes sense that you discuss over the counter antibiotic sales as that is a contributor to inappropriate antibiotic use. However, I feel you could trim this section down a little since it is not the focus of your analysis.

Response : As suggested the section on over the counter antibiotic sales has been trimmed.

Reviewer : Please review for grammar and punctuation and ensure all abbreviations are defined at their first use.

Response: The manuscript has been reviewed and revised to address this comment.

Reviewer : I think it would be fine to say IQVIA instead of IMS Health since that is the current company name.

Response: The manuscript has been reviewed and revised to address this comment.

Reviewer #2: 

Reviewer: This study proposed to estimate outpatient antibiotic prescription rate and patterns in the private sector in India during one year using one medical audit dataset from IMS Health (now IQVIA). The relevance of this study is the increasing room taken by the big data in healthcare sciences and the switch of paradigm that will be linked to this evolution.

This manuscript exposed their automated method to estimate antibiotic prescription rates and patterns of prescription according to coding practice in a sample of medical reports from the outpatient private sector in India. To conclude the study reported high antibiotic prescription level overall, by antibiotic classes, age groups, and ICD-10 mapped disease conditions. 

The topic is highly relevant because antibiotic consumption remains a major public health problem worldwide, promoting the spread of antimicrobial-resistant organisms. Moreover, to measure and survey the antibiotic prescription rate could represent an added value in combination with infection control strategies, for both achieving successful outcomes in patients and impacting the rate of prescriptions.

Although this approach is meriting, the issues of the paper are not meaningful enough and convincing. I do not agree with the conclusion of the authors: data results could not lead to the conclusion “We observed an inappropriate and high antibiotic prescription rates for upper respiratory infections in children age less than 5 years”. Indeed, you used a dataset to estimate the rates and the motif of prescription; to prescribe an antibiotic is associated with a lot of arguments and to estimate if appropriate or not you must read the entire medical report, with biological results (you did not include) the symptoms, the X-rays and so on. To use retrospective medical and clinical data to estimate the rate of AB prescription is a tool we have to develop nowadays, but you can’t conclude on pertinence with this kind of methods.

There are different potential methodological reasons for that, the main being that very different clinical presentations could lead to quote a upper respiratory infections; but if the person has comorbidities or other medical condition, maybe an antibiotic could be necessary. You did not adjust your results on the conditions of the patients. The study could be more relevant in presenting the results without this kind of interpretation that can’t be done. 

Response: Thank you for your detailed and useful comments. I agree, the argument around inappropriate use of antibiotics in any clinical diagnosis has to be based not only on symptoms reported by the patient and captured on the prescription but also on sign of fever, laboratory diagnosis and x-rays. However, in India, majority of patients seek care at primary care physicians in the private sector, where they are prescribed empirical antibiotic therapy without any laboratory investigations, microbiologic testing and culture sensitivity. Multiple studies have highlighted these issues and given the epidemiological and microbiological trends and patterns of acute respiratory infections in India, more often, viral aetiology has been identified in community based studies, which do not usually require empirical antibiotic therapy, hence the argument around inappropriate prescription. However, as per your suggestion, the statement has been removed. 

Specific comments

Reviewer: The introduction is in my opinion too focused on India, comparisons with outpatient antibiotic prescription in other countries (Western countries but more similar Eastern countries).

Response: Thank you for your valuable comments. Your suggestion has been incorporated in the manuscript in the introduction section. The following has been included in the introduction section (paragraph 2, page 3-4) “Although clinical guidelines on judicious antibiotic…….. does help in reducing the frequency of injudicious antibiotic prescribing”

Method section:

Reviewer: it is well written even if some typos remain (cf. manuscript)

Concerning the representativity of your study population, this sample of GP in the private sector in India. 

*how are recruited the GPs? Volontary? At random? Request? Please explain

*how could you be sure they are representative of the entire country?

Response: Details of the sampling strategy have been added in the section ‘Material and Methods’ sub-section ‘Data Source and Setting’ para 1, page 5 of the manuscript. Please see “This data is collected from a panel comprising of 4600 doctors selected through a multi-stage stratified random sampling, which include… »

Further clarification on representativeness has been added in the section ‘Material and Methods’ sub-section ‘Data Source and Setting’ para 1, page 5-6 of the manuscript. Please see « The data is then extrapolated to reflect the prescription pattern of doctors having private practices in towns with population more than a hundred thousand across the country.” 

Reviewer: You use the population census as the denominator; however aren’t you afraid of the part of this population going to the public sector of healthcare? And so you used a higher number of persons than the targeted population, giving a misinterpretation of the ab prescription rates? What do you think?

Response: The proportion of patient visiting the public sector is generally very small in India, especially for outpatient care and therefore the private sector provides a good enough representation of the country in general. We have explained this in the discussion section, para 8 page 15-16 of the manuscript « Therefore, the study is not representative of the prescriptions generated in public sector facilities. This may have resulted in an underestimation of antibiotic prescription rates albeit only marginally, since more than 80% of the population seeks care in private sector and approximately 90% of medicine expenditure occurs in private sector »

We have also added the following as a limitation to the discussions section para 8, page 15. “Since we did not have information on the number of the patients accessing general practices in metros and class 1 and 1a towns, we had to rely on the total population to arrive at the prescription rates. This is another limitation of our study.”

Reviewer: The dataset did not have the motif of the visit and you coded yourself in ICD -10 codes? From what kind of database? Who did choose the motif of consultation? The pattern of the visit?

This is a little surprising as you went to specific data to structured but less specific ones???

Explain the initial data before the transcription in ICD-10 codes.

Response: We used the information provided in the IQVIA medical audit dataset on the doctor’s diagnosis on the prescription as the basis for ICD 10 classification. The following has been included in the section ‘Material and Methods’ sub-section ‘Statistical Analysis’ Para 1, page 6-7 of the manuscript to explain the coding process better:

“The diagnosis was categorized into ICD 10 codes through a search on the online index using specific key words in the diagnosis provided in the dataset which was taken directly from the prescriptions. The idea was to code the ICD 10 codes up to the narrowest (most detailed) level possible depending on the extent of details provided in the diagnosis.”

Reviewer: How did you take into account the fact that the person could come twice in the same week for the same event and first not have Ab than as still ill, has finally been prescribed one treatment.

Response: We were unfortunately not able to take this into account due to the nature of our data, which was not individual level. This is a limitation of our study.

Discussion section: cf manuscript

A bit too long even if interesting. Some extra subject paragraphs that could be taken out in my opinion or shortened.

Response: The manuscript has been reviewed and revised to address this comment.

References:

Some typos and lack of recent literature on that topic. Authors should better screened references databases. 

Response: The manuscript has been reviewed and revised to address this comment.

Some suggestions to help:

Antibiotic Prescribing in Outpatient Children: A Cohort From a Clinical Data Warehouse. Grammatico-Guillon L, Shea K, Jafarzadeh SR, Camelo I, Maakaroun-Vermesse Z, Figueira M, Adams WG, Pelton S. Clin Pediatr (Phila). 2019 Jun;58(6):681-690.

Durkin MJ, Jafarzadeh SR, Hsueh K, Sallah YH, Munshi KD, Henderson RR, et al. Outpatient Antibiotic Prescription Trends in the United States: A National Cohort Study. Infect Control Hosp Epidemiol. 2018 Feb 27;1–6.

Hersh AL, Jackson MA, Hicks LA, the COMMITTEE ON INFECTIOUS DISEASES. Principles of Judicious Antibiotic Prescribing for Upper Respiratory Tract Infections in Pediatrics. PEDIATRICS. 2013 Dec 1;132(6):1146–54.

---

## [Decision Letter · Decision Letter 1]

7 Oct 2019

PONE-D-19-14711R1

Outpatient antibiotic prescription rate and pattern in the private sector in India: Evidence from medical audit data

PLOS ONE

Dear Dr. Farooqui,

Thank you for submitting your manuscript to PLOS ONE. After careful consideration, we feel that it has merit but does not fully meet PLOS ONE’s publication criteria as it currently stands. Therefore, we invite you to submit a revised version of the manuscript that addresses the points raised during the review process.

We would appreciate receiving your revised manuscript by October 31, 2019. To enhance the reproducibility of your results, we recommend that if applicable you deposit your laboratory protocols in protocols.io, where a protocol can be assigned its own identifier (DOI) such that it can be cited independently in the future. For instructions see: http://journals.plos.org/plosone/s/submission-guidelines#loc-laboratory-protocols

We look forward to receiving your revised manuscript.

Kind regards,

Italo Francesco Angelillo, DDS, MPH

Academic Editor

PLOS ONE

Reviewers' comments:

Reviewer's Responses to Questions

**Comments to the Author**

1. If the authors have adequately addressed your comments raised in a previous round of review and you feel that this manuscript is now acceptable for publication, you may indicate that here to bypass the “Comments to the Author” section, enter your conflict of interest statement in the “Confidential to Editor” section, and submit your "Accept" recommendation.

Reviewer #2: All comments have been addressed

Reviewer #3: (No Response)

2. Is the manuscript technically sound, and do the data support the conclusions?

Reviewer #2: Partly

Reviewer #3: Yes

3. Has the statistical analysis been performed appropriately and rigorously? 

Reviewer #2: Yes

Reviewer #3: Yes

4. Have the authors made all data underlying the findings in their manuscript fully available?

Reviewer #2: No

Reviewer #3: Yes

5. Is the manuscript presented in an intelligible fashion and written in standard English?

Reviewer #2: Yes

Reviewer #3: Yes

6. Review Comments to the Author

Reviewer #2: The authors took into account the comments of both reviewers and modified the manuscript accordingly.

This improved the response to the objectives of the study and make the article suitable for publication in my opinion.

Reviewer #3: Many thanks for allowing me to review this revised manuscript.

The authors tried to adequately address the reviewer comments raised in the previous round of review, although they were not able to overcome some shortcomings (e.g. variance estimates, comorbidities and follow-up visits for the same diagnosis were not explored) due to the nature of the study data. However, I believe that inappropriate use of antibiotics is a serious threat to global health, and exploring the pattern of antibiotic prescription among primary care providers represents one of the best tool to promote the appropriateness of use and to control antimicrobial resistance.

The revised manuscript could be approved for publication in PLOS ONE journal, after other minor revision.

I suggest to add in the limitations section that since the dataset did not include the ICD-10 code, the authors coded themselves the ICD-10 starting from the reason of the visit. It may lead to an inaccurate evaluation.

Both the reviewers highlighted lack of recent literature on that topic. All these papers have to be cited and commented: Bianco et al. Infect Drug Resist. 2018;11:2199-2205, Lindberg et al. Scand J Prim Health Care. 2017 Jun;35(2):178-185 (the study findings showed that more than half of the antibiotic prescriptions were dispensed for the diseases of the respiratory system); Davis et al. Antibiotics 2017; 6,4 23, Bert et al. Eur J Public Health. 2017 Jun 1;27(3):506-512 and Napolitano F et al. PLoS One. 2013 Dec 23;8(12):e84177 to emphasize the pivotal role of patients in reducing the inappropriate and excessive utilization of antibiotics.

7. PLOS authors have the option to publish the peer review history of their article (what does this mean?). If published, this will include your full peer review and any attached files.

Reviewer #2: Yes: Dr l Grammatico-Guillon, MD, PhD

Reviewer #3: No

---

## [Author Response · Author response to Decision Letter 1]

20 Oct 2019

Reviewer #2: The authors took into account the comments of both reviewers and modified the manuscript accordingly.

This improved the response to the objectives of the study and make the article suitable for publication in my opinion.

Response: Many thanks for the positive decision on the manuscript. 

Reviewer #3: Many thanks for allowing me to review this revised manuscript.

The authors tried to adequately address the reviewer comments raised in the previous round of review, although they were not able to overcome some shortcomings (e.g. variance estimates, comorbidities and follow-up visits for the same diagnosis were not explored) due to the nature of the study data. However, I believe that inappropriate use of antibiotics is a serious threat to global health, and exploring the pattern of antibiotic prescription among primary care providers represents one of the best tool to promote the appropriateness of use and to control antimicrobial resistance.

The revised manuscript could be approved for publication in PLOS ONE journal, after other minor revision.

Response: Many thanks for your encouraging comments and suggestions. We have incorporated them into our manuscript.

Comment 1: I suggest to add in the limitations section that since the dataset did not include the ICD-10 code, the authors coded themselves the ICD-10 starting from the reason of the visit. It may lead to an inaccurate evaluation.

Response: The suggested limitation has been included in the manuscript in the Discussion section.

Comment 2: Both the reviewers highlighted lack of recent literature on that topic. All these papers have to be cited and commented: Bianco et al. Infect Drug Resist. 2018;11:2199-2205, Lindberg et al. Scand J Prim Health Care. 2017 Jun;35(2):178-185 (the study findings showed that more than half of the antibiotic prescriptions were dispensed for the diseases of the respiratory system); Davis et al. Antibiotics 2017; 6,4 23, Bert et al. Eur J Public Health. 2017 Jun 1;27(3):506-512 and Napolitano F et al. PLoS One. 2013 Dec 23;8(12):e84177 to emphasize the pivotal role of patients in reducing the inappropriate and excessive utilization of antibiotics.

Response: The suggested literature has been incorporated into the discussion section of the manuscript.

---

## [Editor Report · Decision Letter 2]

23 Oct 2019

Outpatient antibiotic prescription rate and pattern in the private sector in India: Evidence from medical audit data

PONE-D-19-14711R2

Dear Dr. Farooqui,

We are pleased to inform you that your manuscript has been judged scientifically suitable for publication and will be formally accepted for publication once it complies with all outstanding technical requirements.

With kind regards,

Italo Francesco Angelillo, DDS, MPH

Academic Editor

PLOS ONE
---

## [Editor Report · Acceptance letter]

5 Nov 2019

PONE-D-19-14711R2 

Outpatient antibiotic prescription rate and pattern in the private sector in India: Evidence from medical audit data 

Dear Dr. Farooqui:

I am pleased to inform you that your manuscript has been deemed suitable for publication in PLOS ONE. Congratulations! Your manuscript is now with our production department. 

With kind regards,

on behalf of

Professor Italo Francesco Angelillo 

Academic Editor

PLOS ONE